# Reliability of Retinal Layer Annotation with a Novel, High-Resolution Optical Coherence Tomography Device: A Comparative Study

**DOI:** 10.3390/bioengineering10040438

**Published:** 2023-03-31

**Authors:** Leon von der Emde, Marlene Saßmannshausen, Olivier Morelle, Geena Rennen, Frank G. Holz, Maximilian W. M. Wintergerst, Thomas Ach

**Affiliations:** 1Department of Ophthalmology, University Hospital Bonn, 53127 Bonn, Germany; 2B-IT and Institut for Informatics, Universität Bonn, 53127 Bonn, Germany

**Keywords:** OCT, spectral-domain, age-related macular degeneration, axial resolution, high resolution, segmentation, annotation

## Abstract

Optical coherence tomography (OCT) enables in vivo diagnostics of individual retinal layers in the living human eye. However, improved imaging resolution could aid diagnosis and monitoring of retinal diseases and identify potential new imaging biomarkers. The investigational high-resolution OCT platform (High-Res OCT; 853 nm central wavelength, 3 µm axial-resolution) has an improved axial resolution by shifting the central wavelength and increasing the light source bandwidth compared to a conventional OCT device (880 nm central wavelength, 7 µm axial-resolution). To assess the possible benefit of a higher resolution, we compared the retest reliability of retinal layer annotation from conventional and High-Res OCT, evaluated the use of High-Res OCT in patients with age-related macular degeneration (AMD), and assessed differences of both devices on subjective image quality. Thirty eyes of 30 patients with early/intermediate AMD (iAMD; mean age 75 ± 8 years) and 30 eyes of 30 age-similar subjects without macular changes (62 ± 17 years) underwent identical OCT imaging on both devices. Inter- and intra-reader reliability were analyzed for manual retinal layer annotation using EyeLab. Central OCT B-scans were graded for image quality by two graders and a mean-opinion-score (MOS) was formed and evaluated. Inter- and intra-reader reliability were higher for High-Res OCT (greatest benefit for inter-reader reliability: ganglion cell layer; for intra-reader reliability: retinal nerve fiber layer). High-Res OCT was significantly associated with an improved MOS (MOS 9/8, Z-value = 5.4, *p* < 0.01) mainly due to improved subjective resolution (9/7, Z-Value 6.2, *p* < 0.01). The retinal pigment epithelium drusen complex showed a trend towards improved retest reliability in High-Res OCT in iAMD eyes but without statistical significance. Improved axial resolution of the High-Res OCT benefits retest reliability of retinal layer annotation and improves perceived image quality and resolution. Automated image analysis algorithms could also benefit from the increased image resolution.

## 1. Introduction

Optical coherence tomography (OCT) has become the standard of care in ophthalmology and has revolutionized diagnostics in retinal diseases including age-related macular degeneration (AMD) [1]. OCT allows for non-invasive imaging of individual layers of the retina creating quasi-histological cross-sections of retinal tissue [2]. Since the first routine clinical use of OCT technology, retinal imaging has tremendously improved in terms of axial resolution, image acquisition time, and signal-to-noise ratio [3,4]. Current state-of-the-art devices in clinical practice are spectral-domain (SD)-OCT and swept-source-(SS) OCT applying unique imaging techniques both resulting in a fast and highly depth-resolved OCT image [5]. Both imaging techniques use Fourier domain transformation for image reconstruction. The key difference between the two is that SD-OCT uses a continuous light source that simultaneously emits a broad spectrum of wavelengths and the spectrometer acts as a detector to separate the different wavelengths, whereas the laser in SS-OCT sequentially emits narrow portions of the source spectrum [6]. SS-OCT provides a lower sensitivity decay with increasing depth range and therefore allows to accurately image deeper structures like the choroid. SD-OCT images on the other hand have been shown to provide better contrast of the vitreoretinal interface and superior optical axial resolution of the retina [7].

The investigational High-Res OCT is a novel SD-OCT device (Heidelberg Engineering, Heidelberg, Germany) that generates images with enhanced axial resolution (up to 3 µm instead of >7 µm) [8]. The improved axial resolution is achieved by deploying a stronger light source (1 mV increased power at the pupil entrance), by increasing the bandwidth by a ratio of more than three, and by using a shorter central wavelength (853 nm instead of the 880 nm) as compared to conventional SD-OCT.

The imaging technique aims to achieve clearer and more detailed images than previous OCT devices. However, so far, no study has investigated if the improved axial resolution actually leads to increased visibility of anatomic details. Additionally, it is unclear if the identification and, hence, annotation of specific retinal layers is actually improved.

Despite the aforementioned strengths of OCT, increased image resolution could help to further improve this widely established and highly relevant examination technique. This might be particularly relevant in degenerative retinal diseases such as AMD. A large-scale study on AMD comparing the inter-reader reliability of new OCT criteria found only moderate inter-grader agreement of some biomarkers (e.g., choroidal hyper-transmission with 0.63 [Gwet’s First-Order Agreement Coefficient (AC1)]; retinal pigment epithelium (RPE) disruption with 0.26 AC1) [9]. Experienced graders stated that, for example, the edges of RPE disruption were difficult to identify on the images. This could be simplified with increased resolution and contrast [10]. Increased image resolution could also be used to refine annotation of retinal layers as well as retinal layers’ thicknesses to improve monitoring of disease progression [11]. Refined detail of annotation could boost drusen volume and photoreceptor thinning analysis in AMD, both important novel OCT biomarkers [12,13,14,15]. Finally, increased image resolution could be used to identify new and so far undetected structural biomarkers and improve the diagnosis and monitoring of retinal diseases including AMD.

Therefore, we determined the intra- and inter-reader agreement of retinal layer annotations on an investigational High-Res OCT device and a conventional SD-OCT. Further, we investigated differences in subjective image quality of these two devices. Our purpose is a first proof of principle study to demonstrate superiority in terms of retinal layer identification and image quality of the High-Res OCT.

## 2. Methods

### 2.1. Subjects

Subjects with AMD were recruited from the Department of Ophthalmology, University of Bonn, Germany. Inclusion criteria were at least 50 years of age, no prior intraocular surgery of the eye except for cataract surgery and the presence of AMD according to the Beckman classification [16]. Exclusion criteria included refractive errors ≥5.00 Diopters of spherical equivalent as assessed by autorefraction (ARK-560A; Nidek, Gamagori, Japan), any signs of current or previous history of ocular diseases, as well as a history of glaucoma or relevant anterior segment diseases with media opacities. If both eyes met the inclusion criteria, the eye with the better visual acuity was selected. Age-matched healthy subjects without any signs for current or previous history of ocular diseases served as controls. Further, control and AMD subjects with systemic diseases such as diabetes were not included in this study. The Ethics committee of the University of Bonn approved the study (application nr. 305/21). All participants gave written informed consent after explaining the nature of this study. All study procedures adhered to the Tenets of the Declaration of Helsinki.

### 2.2. Device Specifications

The investigational High-Res OCT is based on the Spectralis technology (Heidelberg Engineering GmbH, Heidelberg, Germany) with increased optical axial resolution (<3 µm instead of <7 µm as in conventional SD-OCT) by an increased spectral bandwidth (137 nm instead of 50 nm) of the super luminescent diode-based light source, higher OCT power at the pupil (2.2 mW instead of 1.2 mW) and a shift in the central wavelength (880 nm instead of 853 nm). Apart from improved axial resolution, improved OCT power could result in an improved signal-to-noise ratio.

### 2.3. Imaging Protocol

For retinal imaging, a standardized imaging protocol was performed in subjects and controls after pupil dilation with 1.0% tropicamide and 2.5% phenylephrine. Simultaneous confocal scanning laser ophthalmoscopy and SD-OCT imaging was performed using the Spectralis HRA+OCT (Heidelberg Engineering, Heidelberg, Germany): image size 30° × 25°, centered on the fovea, 121 horizontal B-scans (image averaging (ART) 25 frames) and one horizontal central line scan (ART, 100 frames). The same imaging protocol was performed with the investigational High-Res OCT device. Imaging with the two devices was performed in a random order by the same operator.

### 2.4. Image Layer Annotation

Two masked expert medical graders with computer assistance manually annotated retinal layers of all conventional OCT and corresponding central High-Res OCT B-scans in a random order using EyeLab v0.4.2 (bibliography information can be found at https://zenodo.org/record/6816137 (accessed on 1 August 2022), DOI: 10.5281/zenodo.6402100) after export of the images in the raw image format. Saved annotations were imported using Eyepy (DOI: 10.5281/zenodo.7081330) and further processed in Python (https://python.org (accessed on 1 August 2022)). In both modalities, layer heights from A-scans were included in downstream analysis if annotations for all layers were available. Twenty A-scans from both ends of each B-scan were excluded as to not include the optic nerve head (CSV/EXCEL sheets as supplements of all evaluated layer heights and positions are available upon reasonable request).

The layer definitions were slightly adapted from previous publications (Figure 1) [15,17,18,19]. We defined the retinal layers as following. Retinal nerve fiber layer (RNFL): between the internal limiting membrane (ILM) and the lower bounds of the RNFL; ganglion cell layer (GCL): between the RNFL and the lower bound of the GCL; inner plexiform layer (IPL): between the GCL and the lower bound of the IPL; inner nuclear layer (INL): between the IPL and the lower bound of the INL; outer plexiform layer (OPL): between the INL and the lower bound of the OPL; outer nuclear layer (ONL): between the OPL and the external limiting membrane (ELM, Henle fiber layer was included in the ONL in analogy to Sadigh et al. [20]); ELM: between the ELM to the ellipsoid zone (EZ,); EZ: between the EZ and the interdigitation zone (IZ); IZ: between the IZ to the retinal pigment epithelium (RPE); RPE: between the RPE and the Bruch’s membrane (BM); choroid: between the BM and the choroid/sclera interface. Finally, in AMD eyes, the RPE drusen complex (RPEDC) that conjoins the RPE/IZ and encompasses all drusen material, whether below the RPE (soft drusen and cuticular drusen) or between RPE and photoreceptors (subretinal drusenoid deposits (SDDs)) and vitelliform debris) was determined [18,21].

### 2.5. Image Quality Assessment

A subjectively perceived image quality assessment of OCT B-scans using an ordinal qualitative grading scale (1 low quality–10 high quality) in a masked fashion was performed. Two readers graded OCT B-scans for perceived image noise, contrast, resolution and illumination in a random order. A mean opinion score (MOS) of all criteria and from both readers was computed. Of note, images were graded using the visualization display (1:3 µm). As High-Res OCT images were visually distinguishable from the conventional OCT device due to improved image quality, grading has to be considered non-blinded.

### 2.6. Statistical Analysis

Statistical analysis was performed in Python. For retinal annotation lines, the mean absolute error (MAE) was calculated, and confidence intervals (CI) were generated. Additionally, the root mean squared error as an alternate metric was calculated (Appendix A). Finally, a t-test was performed to compare inter- and intra-reader variability between conventional and High-Res OCT for both AMD subjects and healthy controls. The subjectively assessed quality assessment of the graders from the two examined OCT devices was assessed using a Wilcoxon signed rank test. A *p*-value of less than 0.05 was considered statistically significant.

## 3. Results

### 3.1. Demographics

A total of 30 eyes of 30 patients with early/intermediate AMD (mean age ± standard deviation (SD), 75 ± 8 years) and 30 eyes of 30 controls (62 ± 17 years) were included in the study (Table 1). Twenty-one AMD patients exhibited large drusen and/or seven subretinal drusenoid deposits (SDDs) and six patients exhibited pigment abnormalities. None of the control subjects showed any drusen or SDD.

### 3.2. Retest Reliability of Retinal Layer Annotation

Intra- and inter-reader reliability were higher on High-Res OCT (Figure 2). This could be confirmed with a reduced MAE for intra-reader analysis of retinal layer heights for all assessed retinal layers (except the choriocapillaris layer in the AMD group) (Table 2). Statistical significance was reached in the control group for the: RPE, EZ, ELM, OPL, INL, GCL and RNFL. Similarly, for AMD eyes, statistical significance for improved retest reliability of layer annotation for the High-Res OCT was reached for ELM, IPL and RNFL. When comparing overall intra- and inter-reader variability (both devices), Bland–Altman plots revealed reduced inter-reader reliability for High-Res OCT measurements (Figure 2). Nonetheless, High-Res OCT also showed increased inter-reader retest reliability in most retinal layers compared to the conventional OCT (Table 3): the inner retina showed highest noticeable differences with statistically significant improved retest reliability of EZ, GCL and RNFL layers in controls. In AMD, the OPL, IPL and GCL and RNFL proved to have statistically significant better retest reliability of layer annotation of the High-Res OCT. Interestingly, both the intra- and inter-reader agreement of the choriocapillaris were statistically significantly higher (*p* = 0.02) in conventional OCT imaging (unlike all other layers). In AMD eyes, there was no statistical difference between High-Res OCT and conventional OCT for the RPEDC layer.

### 3.3. Image Quality Assessment

The overall MOS for High-Res OCT and conventional OCT were 9 and 8, respectively, with statistically significant improved image quality for the High-Res OCT (Z-value = 5.4, *p* < 0.01). For individually graded criteria, most noticeable differences were present in perceived spatial resolution with a mean of 9 or 7 (High-Res OCT/conventional OCT), reaching statistical significance (Z-value = 6.2, *p* < 0.01). There was no statistically significant difference for the other criteria (perceived image noise, contrast and illumination) between High-Res OCT and conventional OCT.

## 4. Discussion

This study provides a detailed analysis of retinal layer annotation accuracies and image quality for High-Res OCT and conventional OCT in health and disease. We showed improved intra- and inter-reader retest reliability of most retinal layer annotations for the High-Res OCT as compared with a conventional OCT device. Further, our results indicate that the perceived improved image quality mainly derives from improved image resolution. To the best of our knowledge, this is the first study evaluating the effect of the novel High-Res OCT technique on retinal layer annotation reliability and subjective image quality.

In line with other publications, we found a disparity between inter- and intra-reader reliability for retinal layer annotations [22,23,24]. As expected, these were most pronounced in layers difficult to delineate, such as the outer plexiform layer where graders had to account for the Henle Nerve Fiber dispersion [25]. Further, retinal layers that were difficult to identify such as the interdigitation zone in AMD (see below) revealed slight bias of the individual graders over the intra-reader agreement [26,27]. Nonetheless, both intra- and inter-reader reliability demonstrated similar trends of improved retest reliability and identified similar retinal layer annotation accuracies indicating a strong advantage of High-Res OCT. The agreement of inter- and intra-reader reliability substantiates the benefit of this novel device.

A notable finding of the present study was that High-Res OCT significantly improved retest reliability in the inner retina. Analyzing the High-Res OCT image, we expected most notable differences in the outer retina. Interestingly, automated annotation algorithms developed by different research groups are also most accurate in the inner retina and especially at the RNFL that has been evaluated extensively and shows excellent retest reliability (e.g., interclass correlation coefficients between 97–99%) [15,28,29,30]. Additionally, our cohort of AMD and healthy controls could have slanted better results in the inner compared to the outer retina. Albeit inner retinal thinning in association with AMD has been described, most noticeable differences are found in the outer retina in this disease [31,32]. The finding of improved retest reliability for the inner retina using High-Res OCT nonetheless warrants further investigation and might have clinical implications also in other retinal diseases. Changes in RNFL layer thickness provide an opportunity to commence or increase treatment before significant decline in vision in glaucoma [33,34,35]. Especially, detecting these structural changes over time may even be more advantageous than the comparison to a normative database. High-Res OCT with improved axial resolution could reduce misclassification of disease progression particularly in already thinned nerve fiber layers, e.g., glaucoma. Albeit small, these changes observed with better resolution could make a difference. It was suggested that a short-term change in average RNFL thickness of 4 μm may be considered as suspicious for glaucoma progression, which was similar to the change of 5 μm suggested by Leung et al [36]. Apart from the mentioned RNFL, more accurate GCL and IPL layer thickness measurements of the macula could also be deployed for glaucoma detection and progression as these are often involved in early glaucomatous processes [37,38].

Retinal layer annotation accuracies of the outer retina with the High-Res OCT also showed a trend towards improved retest reliability over the conventional OCT but only proving statistically significant in a limited number of scenarios (e.g., RPE layer control group intra-reader annotation, EZ layer both groups). The graders reported that the identification of the interdigitation zone in the control group with the High-Res OCT device was impeded as it often appeared to split up into two different retinal layers. This observation is in line with findings from adaptive optics OCT [39]: photoreceptor related layers between the ellipsoid zone and the RPE are probably split up into the cone outer segment tip (COST) and the rod outer segment tips (ROST). As shown in Figure 1**,** splitting cannot be observed at the fovea where rod photoreceptors are absent. Therefore, the definition of the IZ may be too imprecise for High-Res OCT. This finding could also entail clinical consequences as this would allow to quantify rods and cones separately. Potential applications for this new biomarker would be rod/cone dystrophies but also diseases like AMD that show a stronger rod than cone vulnerability [15,18,19]. However, this finding of a split of the interdigitation zone needs further validation using other high-resolution imaging modalities combined with histologic studies.

The presence of AMD hampers the delineation of the IZ. To account for this, in this and past studies we deployed the RPEDC definition (including both drusen material above and below the RPE) [15,18,19]. High-Res OCT was not statistically different in delineating the RPEDC, so we cannot expect superiority for drusen volume determination based on the current data. Further, the photoreceptor layers in most cases were not delineated with statistically significant higher accuracy with the use of High-Res OCT. Future studies are needed to determine if High-Res OCT can in fact help in identifying photoreceptor thinning (novel marker for AMD disease progression) and drusen volume measurements [15]. Further studies should also corroborate if improved accuracy of retinal layer identification translates into more accurate annotations of AMD including biomarkers such as SDD, HRF or beginning atrophy (incomplete RPE and outer retinal atrophy, iRORA) [9,12,13,14,15]. Figure 3 shows three comparative OCT scans of both the High-Res OCT and conventional OCT. Some retinal layers appear to be more clearly distinguishable and less blurry. Clinical translation needs to be addressed in further detail in future comparative studies. Further, the detailed histopathological–clinical correlation of the better visualized retinal structures (e.g., hyperreflective spots in the outer and inner nuclear layer) is warranted.

The inter- and intra-reader agreement of choriocapillaris annotation was poor for the High-Res OCT and thus other OCT modes should be considered when assessing this layer. Future studies should assess accuracy of the enhanced depth imaging modus (EDI) of the High-Res OCT [40]. A comparison to the swept-source OCT would be of further interest [41].

As hypothesized, the MOS and spatial resolution were improved in High-Res OCT. In ophthalmological imaging (e.g., OCT and OCT-angiography), image quality assessment is already routinely used in clinical studies [42]. Most metrics for image quality assessment in image processing applications rely on a sensitivity-based framework (e.g., peak signal-to-noise ratio) [43]. Human-based opinion scores have the advantage to classify image quality more accurately in the presence of pathology and to better assess image quality that is essential for human-based grading [43]. On the other hand, it is more prone to human error, less reproducible and might include readers’ bias for a specific device. In future studies, we are aiming to develop objective image quality metrics that correlate with perceived quality measurement. We assumed that contrast of images would also be statistically significantly better since High-Res OCT images show superior laser power. However, this was only the case for one of the two human graders. A MOS composed of more than two graders might be beneficial for future studies.

Limitations of this study include the use of central B-scans only. Furthermore, for a more detailed analysis, inclusion of comparison of additional OCT modalities would be desirable (e.g., enhanced depth imaging [EDI] or high-resolution [improving lateral resolution] mode). Furthermore, interpretation of our results is limited to healthy eyes and AMD, as we did not include other diseases. Finally, healthy and AMD-affected subjects were in the same age range but not age-matched. This could have further underscored improved image quality and annotation accuracies in the younger healthy participants.

Strengths of this study include that both intra- and inter-retest reliability were evaluated to diminish inter-individual biases of graders. Further, the uniform study protocol for both conventional and High-Res OCT allowed for a fair comparison of both devices. Additionally, analyzing image layer annotations and image quality in both health and disease enabled us to evaluate the devices in a clinically relevant scenario.

In summary, we demonstrated that the High-Res OCT has the potential to improve identification of retinal layers in health and disease and the annotation of imaging biomarkers in degenerative retinal diseases such as AMD. Further, High-Res OCT allows for improved visualization and stratification of anatomical details including COST and ROST in the interdigitation zone. The improved image quality and axial resolution may allow to further elucidate the pathophysiology of retinal diseases like AMD and improve clinical–histological comparisons in routine clinical practice.

## Figures and Tables

**Figure 1 bioengineering-10-00438-f001:**
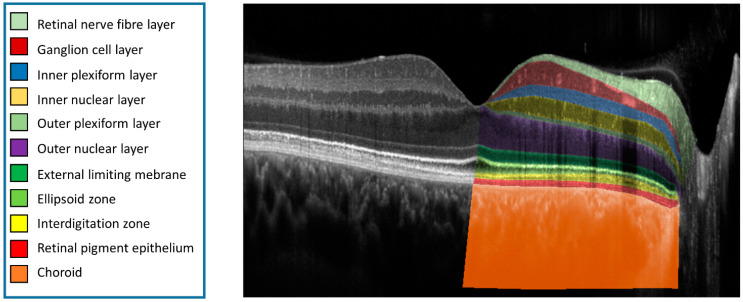
OCT Layer Annotation. Eleven different layers (from retina and choroid), annotated and color-coded on a central B-scan from a High-Res OCT image. The right eye of a 28-year-old male control participant (best corrected visual acuity of 1.0).

**Figure 2 bioengineering-10-00438-f002:**
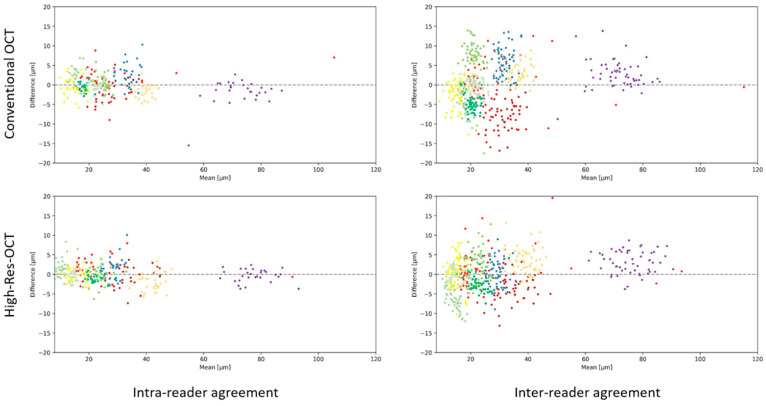
Bland–Altman Plots of Intra- and Inter-Reader Agreement. The x-Axis shows the mean retinal layer thickness in micrometer (µm). The y-axis shows the differences in retinal layer thicknesses (in µm) between duplicate grading by one reader (intra-reader, left column) and between two readers (inter-reader, right column) for the conventional Spectralis OCT (upper row) and for the High-Res OCT (lower row) device. Each color-coded dot represents a specific retinal layer of a participant. Color codes are elucidated in the legend on the top right corner.

**Figure 3 bioengineering-10-00438-f003:**
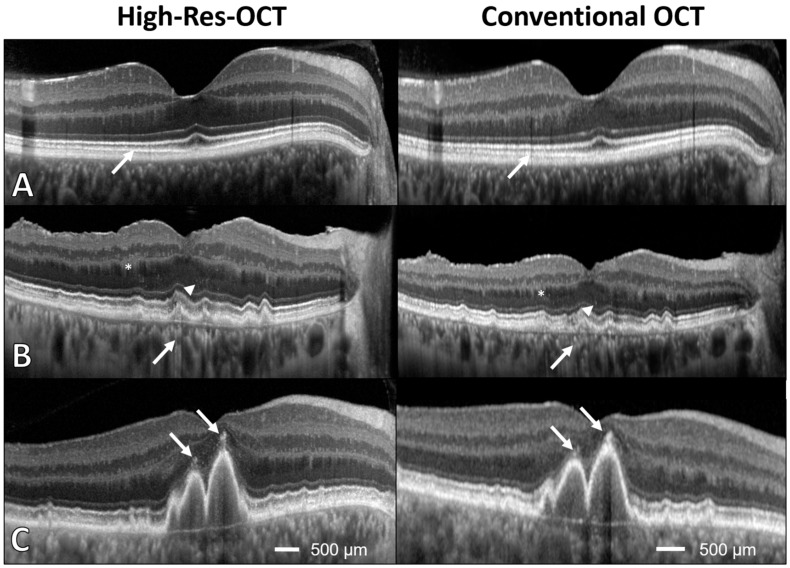
Comparison of conventional and High-Res OCT: healthy eye and exemplary AMD lesions. B-scans of AMD typical lesions for both the High-Res OCT and the conventional OCT. (**A**): High-resolution OCT (left) and conventional spectral-domain OCT (right) of a right eye from a 60-year-old healthy control (best corrected visual acuity logMAR 1.0) for comparison. Note the improved delineation of the interdigitation zone in the High-Res OCT highlighted by the white arrows. (**B**): Right eye of a 67-year-old participant (best corrected visual acuity logMAR 0.8) with presence of sub-RPE drusen and with a small lesion of focal RPE attenuation and underlying choroidal hyper-transmission only detectable in the High-Res OCT (white arrow). In addition, in the High-Res OCT the inner retinal layers appear as less blurry and hyperreflective layers as the Henle’s Fiber layer (asterisk) or hyperreflective spots (arrow head) within the outer nuclear layer are better visualized. (**C**): Left eye of a 72-year-old participant with sub-RPE drusen and hyperreflective foci (HRF; best corrected visual acuity logMAR 0.63). In the conventional SD-OCT image, RPE still appear to be attached to the underlying RPE layer, while at the same position in the High-Res OCT, HRF appear more distinguishably separated from the underlying RPE layer.

**Table 1 bioengineering-10-00438-t001:** Study cohort‘s characteristics.

	AMD	Controls
Participants	30	30
Age (years ± SD)	75 ± 8	62 ± 17
Sex female	20	12
Laterality right eye	13	16
Visual acuity (mean ± SD)	0.2 ± 0.11 logMAR	0.03 ± 0.05 logMAR

**Table 2 bioengineering-10-00438-t002:** Retinal layer annotation accuracy: intra-reader reliability.

		Mean Absolute Error of Retinal Layer Thickness Annotations [µm]											
Modality	Group	CHO	BM	RPE	IZ	EZ	ELM	OPL	INL	IPL	GCL	RNFL	ILM
High-Res OCT	AMD	19.8 [4, 35]	3.1 [2.3, 3.9]	3.5 [2.8, 4.3]	3.8 [3, 4.6]	2.2 [1.9, 2.6]	**1.7 *** **[1.5, 2]**	2.9 [2.4, 3.6]	2.5 [2.1, 2.9]	**2.9 *** **[2.5, 3.2]**	3.9 [3, 4.9]	**2.5 *** **[2.1, 2.9]**	1 [0.1, 1.9]
	Control	17.6 [11, 23]	1.3 [0.4, 2.2]	**2.2 *** **[1.6, 2.9]**	2.5 [1.9, 3]	**1.3 *** **[1.1, 1.5]**	**1 *** **[0.8,1.2]**	**2.3 *** **[2.1, 2.6]**	**2 *** **[1.5, 2.4]**	**2.4 *** **[2.1, 2.7]**	**3 *** **[2.5, 3.6]**	**2.3 *** **[1.9, 2.6]**	0.1 [0, 0.5]
Conventional_ OCT	AMD	13.2 [6, 20]	3.9 [2.2, 5.6]	3.8 [3.2, 4.4]	4.1 [3.3, 5]	2.5 [2, 3.1]	2.5 [1.8, 3.2]	3.9 [2.4, 5.3]	2.8 [2.3, 3.3]	3.7 [3.1, 4.3]	4.3 [3.7, 5]	2.9 [2.5, 3.3]	0.1 [0, 0.3]
	Control	21.1 [13, 28]	1.3 [0.2, 2.4]	3 [2.6, 3.4]	2.6 [2.3, 2.8]	1.7 [1.5, 1.9]	1.6 [1.5, 1.7]	3.1 [2.6, 3.6]	2.8 [2.4, 3.2]	3.2 [2.9, 3.5]	4.7 [3.9, 4.5]	2.8 [2.5, 3.1]	0.4 [0, 1.3]

Mean absolute error (MAE) and 95% confidence intervals in brackets of retinal layer thicknesses [µm] between duplicate gradings by one reader. Values marked with an asterisk are significantly smaller in the High-Res OCT data (α = 0.05). Abbreviations: CHO: Choroid; BM: Bruch’s Membrane; RPE: Retinal Pigment Epithelium; IZ: Interdigitation Zone; EZ: Ellipsoid Zone; ELM: External Limiting Membrane; OPL: Outer Plexiform Layer; INL: Inner Nuclear Layer; IPL: Inner Plexiform Layer; GCL: Ganglion Cell Layer; RNFL: Retinal Nerve Fiber Layer; ILM: Internal Limiting Membrane; AMD: Age-Related Macular Degeneration.

**Table 3 bioengineering-10-00438-t003:** Retinal Layer Annotation Accuracy: inter-reader reliability.

		Retinal Layers											
Modality	Group	CHO	BM	RPE	IZ	EZ	ELM	OPL	INL	IPL	GCL	RNFL	ILM
High-Res OCT	AMD	43.5 [29, 58]	3.2 [2.2, 4.2]	5.9 [4.4, 7.3]	5.8 [5.1, 6.5]	**4.3 *** **[3.8, 4.7]**	3.1 [2.6, 3.6]	**5 *** **[4.4, 5.7]**	3.5 [3, 3.9]	**3.7 *** **[3.3, 4.1]**	**5.8 *** **[4.9, 6.8]**	3 [2.6, 3.4]	1.2 [0.4, 2.1]
	Control	54.1 [34, 73]	0.7 [0.3, 1.2]	3.3 [2.8, 3.8]	3.9 [3.2, 4.5]	**4.4 *** **[4.1, 4.9]**	2.5 [2.2, 2.9]	5.5 [4.7, 6.2]	3.5 [2.9, 4.2]	3.9 [3.5, 4.4]	**5.3 *** **[4.6, 6.1]**	**2.7 *** **[2.4, 3]**	0.2 [0, 0.4]
Conventional_ OCT	AMD	38.8 [31, 47]	3.9 [2.7, 5.1]	6.4 [5.1, 7.7]	6.5 [5.6, 7.3]	6 [5.1, 6.9]	3.2 [2.8, 3.7]	6.3 [5.1, 7.5]	3.9 [3.6, 4.3]	4.4 [3.9, 4.8]	9.1 [7.9, 10.2]	3.3 [2.9, 3.6]	0 [0, 0.1]
	Control	38.5 [31, 45]	1 [0.2, 1.9]	3.2 [2.7, 3.7]	3.1 [2.7, 3.5]	6.4 [5.8, 7]	2.5 [2.2, 2.7]	4.4 [3.7, 5.2]	3.4 [3.1, 3.8]	4.2 [3.7, 4.7]	9.5 [8.2, 10.9]	3.1 [2.7, 3.5]	0.3 [0, 0.8]

Mean absolute error (MAE) and 95% confidence intervals in brackets of retinal layer thicknesses [µm] between two readers. Values marked with an asterisk are significantly smaller in the High-Res OCT data (α = 0.05). Abbreviations: CHO: Choroid; BM: Bruch’s Membrane; RPE: Retinal Pigment Epithelium; IZ: Interdigitation Zone; EZ: Ellipsoid Zone; ELM: External Limiting Membrane; OPL: Outer Plexiform Layer; INL: Inner Nuclear Layer; IPL: Inner Plexiform Layer; GCL: Ganglion Cell Layer; RNFL: Retinal Nerve Fiber Layer; ILM: Internal Limiting Membrane; AMD: Age-Related Macular Degeneration.

## Data Availability

The data presented in this study are available on request from the corresponding author. The data are not publicly available due to ethical reasons.

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
