# Peer review of "Reliability of Retinal Layer Annotation with a Novel, High-Resolution Optical Coherence Tomography Device: A Comparative Study"

_bioengineering, 2023, doi:10.3390/bioengineering10040438_

Round 1
Reviewer 1 Report
Emde et al. presented a comparative study of imaging quality and reliability of retinal layer annotation between a high resolution OCT platform and a conventional OCT device. The authors have used systematic experimental controls and solid statistical analysis to demonstrate how improved axial resolution can lead to enhanced visibility of anatomic details in the case of AMD patients. The authors have also fully discussed the contributions, limitations, and implications of the comparative study. I suggest to publish with minor revisions. Please see below for my comments:
• In the Abstract section, Line 6, please replace “Spectralis” with “OCT” since “Spectralis” is a company, not a technology.
• In Figure 2, what is A? Please explain it in the figure caption. Also is it possible to add scale bars in the figure?
• In Figure 3, could the authors add “High-Res-OCT” and “Conventional OCT” on the left, so the readers can easily understand the data without referring to the figure caption? Another question is about the X axis, which shows mean retinal layer thickness. I am just wondering why the measured thickness is in the range of 0.01 to 0.10 um (10 nm to 100 nm), while the axial resolution is ~3 um? Could the authors double check to see whether the unit is mm for both X axis and Y axis?
• Also please unify the citation format in the Reference section.
Author Response
Thank you for your Review. Please see our attached point-by-point file.

Reviewer 2 Report
This manuscript compares conventional OCT with high resolution OCT. This is a very good study but poorly analyzed and written manuscript. There are multiple issues that needs to be addressed before acceptance:
1) The introduction fails to explain what is unique about this paper. It merely says that it compares axial resolution and leaves the onus on the audience to read other references to understand the significance of the paper. There has to be more explanation and more background on the specific topic researched.
2) There is no figure in the introduction and the figure in the abstract frankly does not show much difference between the conventional and high resolution OCT. It looks over-hyped than what the reality is. Th abstract figure has no legend and we don't know what we are looking at.
3) The abstract figure claims to study 30 healthy and 30 AMD; but I could not understand the table 2 & 3 with respect to this information. Where is the data of the 60 patients? Where is the cumulative data? What do the values in bracket mean? I believe it is showing deviation in the data. From the deviation there is not much to choose between the two techniques. They have not conducted any statistical test and even if they do, it is most likely will be insignificant. In that case, how come the authors have favored the high resolution OCT over conventional OCT?
4) Figure 2 should be presented as figure 1 and should be labeled properly. As an audience we sould be able to infer the difference by looking at the data. Honestly, the High Res panel C looks worse than conventional technique and panel D looks same. If that is the best image then there is not much to choose between the two and they look the same.
5) The technique itself has not been explained as how the images are retrieved and no adequate background is provided about the technique.
6) This study is looking at age related macular degeneration and the controls are in the range of 40s, 50s and 60s while AMD patients are in the range of 60s and 70s. There should have been images that compares the 40s to 70 s and specifically compare the 60s healthy vs 60s AMD. Also compare the sexes. Have they verified that patients do not have any other disorder or disease and are not diabetic? This information has not been verified.
7)Intra reader and inter reader has not been explained. I don't get the significance of Fig. 3. But again it does not prove high resolution OCT to the conventional OCT when the data has so much deviation. Please provide analysis of all the analyzed subjects.
8) Laterality, Visual acuity, Large Drusen and SDD; how do they compare from normal vs. AMD? Please explain that in your images.
9) There are no statements on ethics and IRB clearances, acknowledgement and author contributions.
Overall, this is a poorly done study with so much potential. Please heavily revise to be reconsidered.
Author Response

(The authors gave the same response as above.)

Round 2
Reviewer 2 Report
a) The authors still need to add the clearances they got for studying human subjects.Overall, the manuscript looks much better, but the data analysis has still room for improvement. For now, the manuscript is ready to be accepted with the minor addition.
Author Response
Thank you for reviewing our manuscript again and for the kind appreciation of our revised manuscript . Please excuse that we did not copy our clearance into the point-by-point response. Please see our clearance slightly reworded in the manuscript below under the application nr. 305/21.
Changes:
Lines 108-111 now reads: "The Ethics committee of the University of Bonn approved the study (application nr. 305/21). All participants gave written informed consent after explaining the nature of this study. All study procedures adhered to the Tenets of the Declaration of Helsinki."